# Activated Regulatory T-Cells, Dysfunctional and Senescent T-Cells Hinder the Immunity in Pancreatic Cancer

**DOI:** 10.3390/cancers13081776

**Published:** 2021-04-08

**Authors:** Shivan Sivakumar, Enas Abu-Shah, David J. Ahern, Edward H. Arbe-Barnes, Ashwin K. Jainarayanan, Nagina Mangal, Srikanth Reddy, Aniko Rendek, Alistair Easton, Elke Kurz, Michael Silva, Zahir Soonawalla, Lara R. Heij, Rachael Bashford-Rogers, Mark R. Middleton, Michael L. Dustin

**Affiliations:** 1Department of Oncology, University of Oxford, Oxford OX3 7DQ, UK; shivan.sivakumar@oncology.ox.ac.uk (S.S.); alistair.easton@oncology.ox.ac.uk (A.E.); mark.middleton@oncology.ox.ac.uk (M.R.M.); 2Kennedy Institute of Rheumatology, University of Oxford, Oxford OX3 7FY, UK; david.ahern@kennedy.ox.ac.uk (D.J.A.); ashwin.jainarayanan@exeter.ox.ac.uk (A.K.J.); Elke.kurz@kennedy.ox.ac.uk (E.K.); 3Oncology, Oxford University Hospitals NHS Foundation Trust, Oxford OX3 9DU, UK; 4Sir William Dunn School of Pathology, University of Oxford, Oxford OX1 3RE, UK; 5University of Oxford Medical School, Oxford OX1 2JD, UK; edward.arbe-barnes@magd.ox.ac.uk; 6Interdisciplinary Bioscience Doctoral Training Program and Exeter College, University of Oxford, Oxford OX3 7DQ, UK; 7Nuffield Department of Surgical Sciences, University of Oxford, Oxford OX3 9DU, UK; n.mangal@imperial.ac.uk; 8Department of Surgery, Oxford University Hospitals NHS Foundation Trust, Oxford OX3 9DU, UK; srikanth.reddy@ouh.nhs.uk (S.R.); michael.silva@ouh.nhs.uk (M.S.); zahir.soonawalla@ouh.nhs.uk (Z.S.); 9Department of Pathology, Oxford University Hospitals NHS Foundation Trust, Oxford OX3 9DU, UK; Aniko.Rendek@ouh.nhs.uk; 10Department of General, Gastrointestinal, Hepatobiliary and Transplant Surgery, RWTH Aachen University Hospital, 52074 Aachen, Germany; lheij@ukaachen.de; 11Institute of Pathology, University Hospital RWTH Aachen, 52074 Aachen, Germany; 12Wellcome Trust Centre for Human Genomics, University of Oxford, Oxford OX3 7BN, UK; rbr1@well.ox.ac.uk; 13Oxford NIHR Biomedical Research Centre, Oxford University Hospitals NHS Foundation Trust, Oxford OX3 9DU, UK

**Keywords:** pancreatic cancer, immune checkpoints, TIGIT, CD39, ICOS, regulatory T-cells, senescent T-cells

## Abstract

**Simple Summary:**

Pancreatic cancer has the worst survival of any human cancer. Checkpoint blockade has not yielded much benefit in pancreatic cancer. We explored immune cell phenotypes with this disease to identify new targets for checkpoint blockade therapy. We created a checkpoint-focused panel to analyse immune cells from eight pancreatic cancer patients. This showed us the majority of T-cells are senescent. Further T-cell investigation demonstrated the majority of cytotoxic T-cells have intermediate to low PD1 expression suggesting why PD1 may not work as a pancreatic cancer therapy strategy. Our data has also highlighted a regulatory T-cell population which is highly activated and can mediate immunosuppression. The checkpoints that are highly expressed on these cells are TIGIT, ICOS and CD39, suggesting inhibition of these may be a viable therapeutic strategy. Furthermore, we showed that Tregs were retained amongst the fibroblast stroma of the tumour. Our work suggests there are key checkpoints on Tregs that may help guide therapeutic strategies in pancreatic cancer.

**Abstract:**

Pancreatic cancer has one of the worst prognoses of any human malignancy and leukocyte infiltration is a major prognostic marker of the disease. As current immunotherapies confer negligible survival benefits, there is a need to better characterise leukocytes in pancreatic cancer to identify better therapeutic strategies. In this study, we analysed 32 human pancreatic cancer patients from two independent cohorts. A multi-parameter mass-cytometry analysis was performed on 32,000 T-cells from eight patients. Single-cell RNA sequencing dataset analysis was performed on a cohort of 24 patients. Multiplex immunohistochemistry imaging and spatial analysis were performed to map immune infiltration into the tumour microenvironment. Regulatory T-cell populations demonstrated highly immunosuppressive states with high TIGIT, ICOS and CD39 expression. CD8^+^ T-cells were found to be either in senescence or an exhausted state. The exhausted CD8 T-cells had low PD-1 expression but high TIGIT and CD39 expression. These findings were corroborated in an independent pancreatic cancer single-cell RNA dataset. These data suggest that T-cells are major players in the suppressive microenvironment of pancreatic cancer. Our work identifies multiple novel therapeutic targets that should form the basis for rational design of a new generation of clinical trials in pancreatic ductal adenocarcinoma.

## 1. Introduction

Pancreatic ductal adenocarcinoma (PDAC) has one of the worst outcomes of any human cancer with a 5-year survival of around 7% [1,2]. Early diagnosis with surgical resection followed by combination chemotherapy offers the best chance of long-term survival [3]. We and others have shown that the immune infiltrate in the primary pancreatic tumour is prognostic of the clinical course after a surgical resection [4,5]. Pancreatic cancer has a complex immune microenvironment with T-cells, macrophages, neutrophils, NK cells, B-cells and dendritic cells present [4,6,7,8,9]. The immune infiltrate is dependant on the specific transcriptional subtype a patient may have and this can have implications for chemotherapy [10]. 

Checkpoint blockade immunotherapies, especially antibodies to PD-1 and PD-L1, reactivate tumour-specific T-cells and have demonstrably improved the prognosis of melanoma and lung cancer [11,12]. However, these immunotherapies have had minimal effects on outcomes in pancreatic cancer, with no durable responses in patients [13,14,15,16]. There is some suggestion that checkpoint therapy can be enhanced in pancreatic cancer by the additional of adjuncts such as oncolytic viruses [17].

Due to the poor response of checkpoint blockade agents in PDAC, we propose taking a step back and characterising the states and specific populations of T-cells in this disease, which is the focus of this paper. T-cell infiltration has been reported in pancreatic cancer [18,19] and is a potential target for immunomodulation [20]. Regulatory T-cells (Tregs) have previously been shown to be present in the microenvironment of pancreatic. Cancdf and that they may have an effect on CD11c dendritic cells that will inhibit CD8^+^ T-cells [21]. Mouse studies have suggested potential ways of influencing Tregs. An early study showed depleting Tregs in pancreatic cancer helps recruitment of CD8^+^ T-cells [22]. Conversely a study last year in a Kras driven mouse model showed that depleting Tregs sped up carcinogensis [23]. Although it has been shown that CD4^+^, CD8^+^ and Tregs infiltrate the microenvironment of PDAC, little is known about their phenotype, differentiation or activation status.

Furthermore, although cancer therapeutics has been dominated by PD-1 and CTLA-4 targeting antibodies, many other checkpoints with the potential to impinge on the clinical course of PDAC, including TIGIT, Tim3, Lag3 and CD39, have been identified. Co-stimulatory molecules; which promote suppressive function of CD4^+^ Tregs and CD8^+^ suppressor cells, such as ICOS, OX40, CD40L, GITR and 4-1BB may also bear upon outcomes [24,25]. 

In this study we characterised the immune landscape in primary tumours and the periphery of patients with pancreatic cancer, focusing on T-cells’ functional states, and their immune checkpoint expression patterns. 

## 2. Results

### 2.1. Heterogenous and Suppressive Innate Immune Cell Composition within Pancreatic Cancer 

Fresh tumour resection samples and matched blood from patients undergoing surgery (Appendix A) were used to analyse the tumour microenvironment. We identified the main cellular components and immune lineages with a panel of mass-cytometry (CyTOF) antibodies (Figure 1A, Appendix A). The epithelial cell content of samples was 25.81 ± 6.92% (median ± sd) and that of the stroma 50.06 ± 14.66% (Figure 1B, Appendix A), in agreement with the highly fibrotic nature of PDAC [26]. All 8 patients exhibited a significant level of immune cell involvement, with CD45^+^ being 22.85 ± 12.60% of all live cells (Appendix A). 

We identified the main immune cell lineages illustrated in the viSNE plot and marker expression maps (Figure 1C). We further characterised different metaclusters of cells corresponding to different lineages using unsupervised hierarchical clustering (Figure 1D; coloured viSNE and heatmap). There were multiple shared features across all patients including the presence of CD4^+^ T-cells (metacluster 11), CD8^+^ T-cells (metacluster 7), granulocytes (metaclusters 8,10) and mononuclear phagocytes (metaclusters 1,2) and myeloid-derived suppressor cells (MDSCs; metaclusters 4,5). Interestingly, both B-cells (metacluster 3) and NK cells (metacluster 9) numbers were negligible (<0.5%) in some patients’ tumours (Figure 1D, inset). For example, the tumour in patient 3 was deficient in B-cells and that in patient 8 deficient in NK cells (Appendix A). 

We identified multiple subsets of NK cells (Figure 1E, Appendix A), granulocytes (Figure 1F, Appendix A) and mononuclear phagocytes (Figure 1G, Appendix A). Strikingly, most NK metaclusters expressed the inhibitory molecule TIGIT, with those having the highest expression being granzyme B (GzmB) negative, indicative of defective cytotoxicity (e.g metacluster 7). These cells also exhibited tissue residency features, such as expression of the adhesion molecule CD103. Conversely, the infiltrating NK populations had features of cytotoxicity with CD16, CD57 and GzmB expression (metaclusters 4,6). 

The majority of the granulocytes were CD15^+^CD16^+^CD14^±^, corresponding with granulocyte-MDSCs (G-MDSC) [27,28] or antigen-presenting tumour associated neutrophils [29], marked by their HLA-DR expression (metacluster 4), although we also observed the co-expression of PD-L1.

In the myeloid compartment, we identified MDSCs expressing low levels of HLA-DR and high PD-L1 (metaclusters 3), and G-MDSC (metacluster 6). The majority of myeloid cells had an intact antigen presentation capability marked by high levels of HLA-DR (metaclusters 2,4,5). This is in contrast to previous reports describing the major myeloid infiltration in PDAC to be MDSCs [27,30].

Peripheral blood from the same patients presented different population compositions (Appendix A). Specifically, we observed a small percentage (1.26 ± 1.37%) of low-density neutrophils in the PBMC samples (Appendix A, metacluster 11,14) which have been described in PBMCs of patients with tumours [31]. Circulating NK cells, unlike the tumour-associated NK cells, appeared to retain their cytolytic activity with most sub-populations expressing CD56^dim^, CD57, GzmB and CD16 (Appendix A). We also observed a significant population of circulating MDSCs (Appendix A, metacluster 3, 27.82 ± 8.09%) and the presence of T-cell/monocyte conjugates (Appendix A, metaclusters 2 and 13, 0.85 ± 0.42%) expressing PD-L1 (Appendix A, metacluster 1) suggestive of an immune perturbation [32] (Appendix A, metacluster 1). 

In summary, the tumour microenvironment contains diverse populations of immune cells. The main hallmark of these cells is either their non-immunogenecity (low antigen presentation) or ineffective cytotoxicity (TIGIT^+^ NK cells). 

### 2.2. Suppressive and Non-Tumour Responsive T-Cells Predominate PDAC Microenvironment

We hypothesised that the lack of activity of established checkpoint inhibitors such as anti CTLA-4 and anti PD-1 suggest they may not be prominent pathways in PDAC. To test this hypothesis, we re-evaluated T-cells’ states in the tumour microenvironment. We analysed the CD3^+^ T-cells functional states from PDAC tumours using differentiation, activation and checkpoint markers, to characterise the CD8^+^ (Figure 2A, Appendix A), CD4^+^ (Figure 2B, Appendix A) and Treg (Figure 2C, Appendix A) compartments. 

We identified 14 distinct metaclusters of CD8^+^ T-cells. Multiple metaclusters displayed characteristics of a senescent population, CD57^+^CD27^-^CD28^-^ (metaclusters 5,11; 3.89 ± 3.66% of CD8^+^ T cells) or terminally differentiated CD45RA^+^CD27^−/low^CD28^−/low^ (metaclusters 2,7,8; 40.57 ± 18.06%). These cells have been previously described in the context of aging and viral infections and are associated with proliferative senescence and reduced T-cell signalling, whilst maintaining their cytolytic capabilities [33,34,35]. More recently, these have been observed in the context of cancer [36,37], however, this is the first report to describe them in pancreatic cancer. Two suppressive populations with negative immune modulatory effects were identified: (i) a metacluster of FoxP3^+^ CD8^+^ “regulatory” T-cells (metacluster 12), which was only present in 2 out of the 8 patients (Appendix A), and (ii) an exhausted population expressing high levels of multiple inhibitory receptors (metacluster 6; 4.81 ± 8.73%). Interestingly, the exhausted metaclusters exhibited intermediate PD-1 expression, which may explain the limited clinical success targeting the PD1-PDL1 axis (Appendix A). We also identified metaclusters with markers of activation (metaclusters 10,13), proliferation and cytotoxicity (metacluster 9). Overall, we determined that ~ 11.63 ± 8.32% (mean ± s.d) of the CD8^+^ T-cells comprise expression profiles potentiating anti-tumour responses while the majority are either unresponsive (naïve, senescent or exhausted) or even inhibitory. This suggests a certain degree of anti-tumour potential of the CD8^+^ compartment that could be inhibited by other factors preventing the control of the disease. 

We next analysed the contribution from CD4^+^ T-cells as potential mediators to support CD8^+^ T-cells (Figure 2B, Appendix A). The 10 identified metaclusters were divided into three main groups: (i) senescent and terminally differentiated (metaclusters 6,8,4) or non-tumour responsive (metaclusters 2,5,9,10) (~76.15 ± 9.45%), (ii) exhausted (metacluster 7; 2.71 ± 2.03%) and (iii) Foxp3^+^ regulatory (~18.53 ± 8.78%). Indeed, the regulatory CD4^+^ cells comprised between 8.30 and 32.50% of the CD4^+^ population across patients. 

We hypothesised that the Treg population could contribute to inhibiting CD8^+^ T-cell responses. We characterised the precise Treg states within the tumour microenvironment (Figure 2C, Appendix A). The majority of the Tregs (~54.47 ± 19.15%) showed evidence of functional activation and high suppression capacity. They expressed the TNF superfamily receptor 4-1BB, HLA-DR and the inhibitory receptors PD-1 and TIGIT. Some metaclusters showed evidence of cytotoxic activity (CD57^+^, metacluster 2) while the most frequent metaclusters had high expression of TIGIT, ICOS and CD39, indicative of potent inhibitory function [38,39,40,41,42]. 

The peripheral blood CD8^+^ and CD4^+^ T-cells also exhibited a population of senescent T-cells (Appendix A, metacluster 4, 26.02 ± 18.62% of CD8^+^, Appendix A, metacluster 1, 5.21 ± 10.56% of CD4^+^), which might reflect the age of our cohort (median 71 years). Both populations had functionally active subsets; in CD8^+^ an activated and proliferating metacluster was identified (4-1BB^+^HLADR^+^, metaclusters 7,8, 1.12 ± 0.70%). However, there were no exhausted TIGIT^+^ CD8^+^ T-cells in the periphery (Appendix A). There was also a population of cytotoxic CD4^+^ T-cells (GzmB^+^, metaclusters 2,7, 2.61 ± 4.05 %). T-cell/monocyte conjugates, were only observed with CD4^+^ but not CD8^+^ T-cells (CD3^+^CD4^+^CD14^+^, Appendix A, metacluster 11). Finally, the Tregs in the periphery (Appendix A) compromised 9.59 ± 4.30% of the CD4^+^ T-cells, were characterised by high levels of TIGIT and ICOS expression albeit at lower levels compared to their tumour-associated counterparts (~30% higher median expression, Appendix A, respectively). Not surprisingly, there was a larger naïve population (CCR7^+^CD45RA^+^) compared to the tumour, 13.2 ± 5.47% vs. 6.56 ± 3.75%.

In summary, we identified a small proportion of activated CD8^+^ T-cells in the tumour microenvironment. However, the majority of the T-cells are either senescent or terminally differentiated and have little capacity to contribute to the immune response or activated regulatory T-cells that can be suppressing the anti-tumour specific cells. 

### 2.3. Single-Cell RNA Sequencing Validates Senescence and Regulatory Signatures in Tumour Infiltrating T-Cells

The CyTOF analysis highlighted senescent T-cells and TIGIT^+^ICOS^+^ Tregs as potentially important immune cells in pancreatic cancer. We hypothesised similar signatures should be identified on the transcriptional level. We have re-analysed a publicly available single-cell RNA sequencing dataset of 24 PDAC patients from Peng et al. [43] (Figure 3). Focusing on the T-cell compartment, we identified 250 unique clusters as shown in the UMAP (Appendix A, Figure 3A) corresponding to CD8^+^ and CD4^+^ T-cells as well as non-conventional T-cells. 

Among those clusters, we identified 13 Treg clusters based on *FOXP3* expression, all of which exhibit high expression levels of *TIGIT* and co-expressing *ICOS* and *ENTPD-1* (CD39) (Figure 3B, violin plots, Appendix A). We also identified 6 clusters of senescent T-cells (Figure 3C) characterised by increased NK marker expression (*KLRG1*, *KLRB1*) and senescent markers (*HCST*, *HMGB1*) [34], which were significantly elevated in the tumour samples compared to the normal pancreas (Appendix A), suggesting this is a unique characteristic of the tumour microenvironment and is not solely a result of the age of the cohort. Complete differential expression analysis of those populations relative to other CD4^+^ and CD8^+^ T-cells is provided in Appendix A.

Finally, we also identified exhausted cells characterised by the co-expression of at least 3 of the known exhaustion signature genes *PDCD1*, *HAVCR2* (Tim-3), *LAG3*, *TIGIT*, *CTLA4 and ENTPD-1* (Figure 3D). This has captured exhausted clusters with low *PDCD1* expression (Figure 3D violin plot clusters 35, 85, 96, 161). Interestingly, previously reported exhaustion genes such as *TOX* [44], *LAYN* and *MIR155HC* [45], were only upregulated in a subset of the exhausted cell clusters. Together, this suggests a unique exhaustion signature in PDAC-associated T cells.

In summary, we were able to corroborate our observations from the mass cytometry analysis in an independent cohort of PDAC patients at the transcriptomic level. The relative abundance of the T-cell populations identified was similar across the two cohorts (Appendix A).

### 2.4. Effector T-Cells Are Uniformly Distributed within Pancreatic Tumour and Tregs Are Restricted to the Stroma

To investigate the potential cell–cell communication between different T-cell subsets and the surrounding malignant epithelium of the tumour, we analysed cellular spatial distribution using multiplex immunofluorescence (IF) on formalin-fixed paraffin-embedded (FFPE) sections from the same patients as analysed by CyTOF (Figure 4). For each case we identified cancerous, inflamed (pancreatitis) and normal tissue regions, where available (Figure 4A–C, respectively) and annotated the sub-regions into epithelium (based on pan-Cytokeratin staining) and stroma (based on SMA staining). Using the expression of the canonical T-cell markers (CD3^+^, CD4^+^, CD8^+^ and Foxp3^+^), we identified their respective cellular subsets (Appendix A). The CD4^+^ and CD8^+^ distribution within the different regions of the tissue (Figure 4D) was homogeneous with no signs of exclusion from the tumour parencyhma or different stroma regions (Appendix A). Conversely, Tregs were exclusively restricted to the stroma in the cancer and inflamed tissue and almost absent from the epithelium regions. Explaining some recent reports linking Treg depletion to fibroblast pathology in PDAC [23,46]. To further elucidate the relationships between the cells we performed proximity analysis (See methods for details) that revealed the majority of CD8^+^ T-cells to be within 50 mm of the epithelium, with a trend of lower numbers within the cancer region compared to normal albeit not statistically significant (Figure 4E, *p* = 0.1167). 90% of Tregs were in close proximity of a CD8^+^ T-cell, potentially facilitating their immunosuppressive activity across all assessed regions (Figure 4F).

In summary, we note that PDAC microenvironment is well infiltrated with effector CD4^+^ and CD8^+^, suggesting that the lack of anti-tumour response is not related to physical exclusion. On the other hand, the Treg population appears to be restricted to the stroma where it may mediate a more potent suppressive niche. 

## 3. Discussion

Here, we report one of the first comprehensive characterisations of T-cells in primary human pancreatic ductal adenocarcinoma, revealing multiple distinct immune cell signatures of this tumour with potential for informing therapeutic approaches. Previous reports implied the presence of an immunosuppressive tumour microenvironment, but they lacked a detailed definition of its components [16,47]. Here, we identify the different immune cells contributing to this phenotype which include granulocytes and myeloid MDSCs (Figure 1), dysfunctional NK cells and T-cells, and regulatory T-cells (Figure 2). 

We identified clear signatures of dysfunctional effector T-cell populations which are present in both the CD8^+^ and CD4^+^ compartments. The first is an exhausted signature, which surprisingly, is not characterised by the traditional high PD-1 expression, nor does the microenvironment show high expression of its ligand PD-L1 (Appendix A). However, exhausted cells express a different set of inhibitory molecules including TIGIT [48] and CD39 (Figure 2A and Figure 3C). This finding has a direct implication to designing immune-checkpoint trials where we suspect anti PD-1 combinations might have limited, if any, advantage.

Tregs have been shown to be present in the PDAC microenvironment [6], but their functional characteristics are poorly understood. We identified different subtypes of Tregs associated with PDAC, including a highly suppressive Treg population present in all the examined patients (Appendix A), expressing the inhibitory molecule TIGIT and co-stimulatory molecule ICOS (Figure 2C and Figure 3B) whose substantial suppression capacity has been demonstrated by multiple groups [38,40,49]. An interrogation of the expression pattern of the ligands to TIGIT and ICOS has shown distribution across different cellular compartments, which were significantly elevated in the tumour compared to the normal pancreas (Appendix A). There are currently multiple antibodies in development targeting the checkpoint molecules we observed on dysfunctional CD8^+^ T-cells and Tregs and our data calls for trialling them in PDAC. Specifically, anti-TIGIT has been recently proposed as an alternative immunotherapy to anti PD-1 in colorectal, lung and pancreatic cancers [50], anti-CD73 which is the ligand for CD39 has also successfully passed a phase 1 in advanced cancers [51], and anti-ICOS has shown specific targeting of Tregs in mouse models [52]. From our data we can envision these new line of therapeutics having an effect on dysfunctional CD8^+^ T-cells, NK cells and Tregs, eliciting a potent anti-tumour response. 

The spatial analysis of cell distribution in the tumour has revealed a unique localisation of Tregs to the stroma (Figure 4D). The cues that retain the Treg population in the stroma compartment are not obvious and could be a result of the CXCL12-CXCR4 axis [53] or due to the TGFb rich environment maintained by the tumour-associated fibroblasts and contributes to Treg survival [54]. These observations, in light of a recent study in a murine model of PDAC that showed depletion of Tregs to result in disease worsening through an increase in pathogenic fibroblasts [55], highlight the need to understand the details of Treg/fibroblast interaction and its role in disease progression. A possible strategy proposed from this work would be to block exclusively the activity of the ICOS^+^TIGIT^+^ Tregs population while forgoing any depletion approaches that might result in severe adverse effects. 

Finally, we identified a novel senescence signature, which unlike the exhausted phenotype cannot benefit from checkpoint blockade approaches. T-cell senescence has been discussed in the context of viral infections, aging and CAR T-cell therapies and different avenues to replace or rejuvenate those cells through metabolic manipulations, cell therapy and engineering are being investigated [56,57,58]. It would be interesting to understand the mechanism of the observed senescence and whether it is directly linked to the immunosuppressive activity of Tregs in the tumour microenvironment [59,60]. In this a case, we anticipate that lifting Treg suppression will reverse the phenotype while the latter would require interventions to rejuvenate the cells to increase the chances to elicit anti-tumour responses. 

We described features shared between the tumour and the peripheral blood (Appendix A), especially the circulating TIGIT^+^ICOS^+^ Tregs. Those observations made in patients with localised disease raise the possibility of early detection strategies that warrant investigation in larger cohorts. 

## 4. Conclusions

In summary, our data maps the T-cell landscape of pancreatic cancer and we propose multiple novel therapeutic approaches to employ immunotherapies in this recalcitrant disease as well as further scientific investigation. Current pancreatic cancer mouse models appear to lack the immune features we observe using human patient samples [61] highlighting the need to directly test the hypotheses generated from this study and its implication through the design of novel clinical trials. Characterising the immune-landscape in our patient cohort which represents 50% of the total patients diagnosed with pancreatic cancer [62] has a unique advantage of identifying therapeutic approaches with the highest chances of success in patients which are still fit to respond to therapy. 

## 5. Materials and Methods

### 5.1. Patient Recruitment

Samples were collected from 8 patients diagnosed with pancreatic adenocarcinoma (Appendix A) that were fit for palliative operation. The 8 patients consisted of 5 males and 3 females and ranged from ages of 51 to 80. 7 out of 8 patients has adjuvant chemotherapy following the operation, patient 2 and 3 have died within 9 months of the operation while patients 4, 6 and 7 have recurred since. All patients were consented for this study via the Oxford Radcliffe biobank (09/H0606/5+5, project: 18/A031). 

### 5.2. Sample Collection 

From the patients described above, 20 mL blood was collected immediately before surgery into sodium heparin tubes (BD). Tissue samples were placed in RPMI media (Gibco, Waltham, MA, USA) on ice and were reviewed by a designated histopathologist who provided a 10 mm by 10 mm by 3 mm piece for this study. Samples were digested, stained and run on CyTOF. 

### 5.3. PBMC Isolation 

Blood samples were processed within 4 h of collection. 20 mL of 2% FBS/PBS was added to 20 mL of whole blood. This was layered onto Ficoll-Paque. Sample was centrifuged at 1300× *g* for 20 min at the slowest acceleration and with break off. After centrifugation, the PBMC ring was removed using a pipette. The ring was topped up with 2% FBS/PBS and centrifuged again at 300× *g*. Any excess red blood cells were lysed with ACK solution (Life Technologies, A1049201, Waltham, MA, USA) and cells were washed again.

### 5.4. Tissue Digestion 

Sample was initially mechanically disrupted using a scalpel into small pieces. The pieces were put into a 15 mL conical tube, with 9 mL of complete RMPI (10% FBS, 1% Pen/Strep and 1 mM Glutamine) and 1 mL of 10× hyaluronidase/collagenase solution (StemCell, 07912, Vancouver, BC, Canada). A first round of digestion was done at 37 °C for 30 min in a pre-warmed shaker. The supernatant was collected without disrupting the tissue and a fresh digestion media was added (10 mL complete RPMI containing 200 U of collagenase IV (Lorne Laboratories, LS004194, Danhill, Berkshire, UK), 100 mL/mL of DNAaseI (Sigma, DN25, Gillingham, Dorset, UK) and 0.5 U of universal nuclease (Pierce, 88702, Waltham, MA, USA) for an additional 30 min of digestion as before. The supernatant was combined with the one from the first digestion step and the remaining tumour pieces were squeezed through a 100 mm tissue strainer with a further 10 mL of complete RPMI. The supernatants from all digestion steps were combined and centrifuged for 10 min at 300× *g*. Any residual red blood cells were removed with ACK solution.

### 5.5. CyTOF Sample Preparation

Samples were directly taken following isolation for CyTOF staining. Preconjugated antibodies were obtained from Fluidigm or purified antibodies from Biolegend were conjugated in house using Maxpar Conjugation kits (Fluidigm, San Francisco, CA, USA). CD14-Qdot655 was purchased from Thermofisher and acquired in the 114Cd channel-See Appendix A for detailed list of antibodies and clones. Cells were incubated with Intercalator-^103^Rh (Fluidigm, 201103A) for dead cell exclusion, for 10 min at room temperature, followed by staining for surface markers for 20 min at room temperature. Cell fixation and permeabilization was performed using the Foxp3+/Transcription factor staining set (eBioscience, 00-5523-00, Waltham, MA, USA). The nuclear staining protocol was used for the simultaneous detection of cytoplasmic and nuclear targets (Ki67, CTLA-4, granzyme B and Foxp3), staining was done for 20 min at room temperature. An additional fixation step with 1.6% paraformaldehyde diluted in PBS for 10 min at room temperature. The cells were washed and incubated with 0.125 nM Intercalator-191Ir (Fluidigm, 201192A) diluted in Maxpar fix and perm buffer overnight at 4 °C until acquisition. 

### 5.6. CyTOF Data Acquisition 

Immediately prior to acquisition, samples were washed twice with Maxpar cell staining buffer (Fluidigm), twice with cell acquisition solution (Fluidigm) and then resuspended at a concentration of 0.5 million cells/mL in cell acquisition solution containing a 1/10 dilution of EQ 4 Element Beads (Fluidigm, 201078). The samples were acquired on a CyTOF Helios mass cytometer at an event rate of <300 events/second. After acquisition, the data were normalized using bead-based normalization in the CyTOF software. Data were exported as FCS files for downstream analysis. The data were gated to exclude residual normalization beads, debris, dead cells and doublets, leaving DNA^+^ Rh^low^ events for subsequent clustering and high dimensional analyses. 

### 5.7. CyTOF Data Analysis 

Dimensionality reduction visualisation with viSNE and clustering with FlowSOM were done using built in functions in cytobank (https://www.cytobank.org, accessed on 27 February 2021) The number of clusters and metaclusters for the FlowSOM algorithm were reviewed by the researchers. Data was initially overclusterd to identify small populations (all data shown in Appendix A), but for clarity metaclusters were manually combined following researchers’ evaluation and presented in main figures. Heatmaps of normalized marker expression, relative marker expression, and relative difference of population frequency were generated by cytobank and plotted using Prism (GraphPad). Dendograms showing hierarchical clustering of the heatmaps was performed using Morpheus from the Broad Institute (https://www.broadinstitute.org/cancer/software/morpheus/, accessed on 27 February 2021), as an average with 1- Pearson correlation as a parameter. 

### 5.8. Collection of Histological Sections 

Sections were cut on a Leica RM2235 at around 5 microns thickness, floated on a warm water bath, dissected using forceps to isolate the region of interest and lifted centrally onto TOMO slides (VWR, TOMO^®^ 631-1128, Lutterworth, Leicestershire, UK). Sections were air-dried. Samples were sequentially labelled with CD4, CD8, Foxp3, Pan Cytokeratin, and aSMA.

### 5.9. Multiplex Immunohistochemistry 

Multiplex (MP) immunofluorescence (IF) staining was carried out on 4 um thick formalin fixed paraffin embedded (FFPE) sections using the OPAL™ protocol (AKOYA Biosciences, Marlborough, MA, USA) on the Leica BOND RXm autostainer (Leica, Microsystems, Milton Keynes, UK).

Six consecutive staining cycles were performed using the following 1ry Antibody-Opal fluorophore pairings: CD4 (clone 4B12, NCL-L-CD4-368 (Leica Novocastra, Linford Wood, Milton Keynes, UK)–Opal 520); CD8 (clone C8/144B, M7103 (DAKO Agilent, Santa Clara, CA, USA) -Opal 570); CD3 (clone LN10, NCL-L-CD3-565 (Leica Novocastra)–Opal 540); FOXP3 (clone 236A/E7, ab20034 (Abcam, Cambridge, UK)–Opal 620); Pan Cytokeratin (clone AE1/AE3, M3515 (DAKO Agilent)–Opal 650) and aSMA (rabbit polyclonal, ab5694 (Abcam) -Opal 690). 

Primary (1ry) Antibodies were incubated for one hour and detected using the BOND™ Polymer Refine Detection System (DS9800, Leica Biosystems, Milton Keynes, UK) as per manufacturer’s instructions, substituting the DAB for the Opal fluorophores, with a 10 min incubation time and without the Haematoxylin step. Antigen retrieval at 100 °C for 20 min, as per standard Leica protocol, with Epitope Retrieval (ER) Solution 2 (AR9640, Leica Biosystems) was performed before each 1ry antibody was applied. Slides were then mounted with VECTASHIELD^®^ Vibrance™ Antifade Mounting Medium with DAPI (H-1800-10, Vector Laboratories, Burlingame, CA, USA). Whole slide scans and multispectral images (MSI) were obtained on the AKOYA Biosciences Vectra^®^ Polaris™. Batch analysis of the MSIs from each case was performed with the inForm 2.4.8 software provided. Finally, batched analysed MSIs were fused in HALO (Indica Labs), to produce a spectrally unmixed reconstructed whole tissue image, ready for analysis.

Cover slips were lifted post multiplex staining and CD68 (Clone PG-M1, Dako M0876) antibody was stained for chromogenically on the Leica BOND autostainer. Antigen retrieval at 100 °C for 20 min with Epitope Retrieval Solution 2 (AR9640, Leica Biosystems); primary antibody incubation at 1/400 dilution for 30 min then detection using the BOND™ Polymer Refine Detection System (DS9800, Leica Biosystems) as per manufacturer’s instructions.

### 5.10. Multiplex Immunohistochemistry-Image Analysis 

Scanned slides were analysed using Indica Labs HALO^®^ (version 3.0.311.407, Albuquerque, NM, USA) image analysis software. Multiplex and brightfield images were manually annotated by a pathologist, defining areas of pancreas, pancreatitis, pancreatic adenocarcinoma and lymph node. The pathologist taught an integrated Random Forrest Classifier module to segment the multiplex images into stroma and epithelium, with obvious areas artefactual staining manually excluded. A separate Random Forest Classifier algorithm was taught to segment tissue into areas of high, medium and low smooth muscle actin (aSMA) expression. Analysis and cell detection/phenotyping was done using Indica Labs–HighPlex FL v3.1.0 (fluorescent images, Albuquerque, NM, USA) and Indica Labs–Multiplex IHC v2.1.1 (brightfield images, Albuquerque, NM, USA). Cells were annotated based on their marker expression as follows: Epithelium (DAPI+ Cytokeratin+), CD4 helper (DAPI+CD4+), CD8 cytotoxic (DAPI+CD8+) and regulatory T-cell (DAPI+CD4+Foxp3+). Multiplex and brightfield images were registered and topological analysis was carried out using integrated proximity analysis modules. Proximity analysis was done using a 50 mm with 20 bands cut-off as this allowed us to capture physically interacting cells (within ~20 mm radius) as well as account for cells that could contribute to soluble effector molecule gradients. Statistical analysis was done using 2-way ANOVA in Prism (GraphPad, San Diego, CA, USA).

### 5.11. Single-Cell RNA Sequencing Analysis: Pre-Processing, Integration and Batch Correction

FastQ files for 24 PDAC and 11 normal samples were downloaded from the Genome Sequence Archive (https://bigd.big.ac.cn/search?dbId=gsa&q=CRA001160, accessed on 27 February 2021), count matrices were generated in Cell Ranger 3.1.0 as per the original paper [43]. Raw count matrices were imported into the Seurat R package and merged [63]. Cells with <200 and >2.5 × 10^10^ genes, <400 and >1 × 10^16^ molecules, and >25% mitochondrial genes were excluded. Batch correction was performed in Harmony [64].

### 5.12. Single-Cell RNA Sequencing Analysis: Single Cell Clustering and Annotation 

Uniform manifold approximation and projection (UMAP) was performed on the scRNAseq harmonised cell embeddings, upon which clustering was performed. 12 broad cell clusters were identified using reference pancreas and immune gene lists (Appendix A). The T-cell cluster was subsetted into a new Seurat object, and UMAP was re-performed using genes relevant to T-cells to generate 250 clusters (Appendix A). 

Mean and 75th percentile normalised count matrices were generated for these clusters. 75th percentile normalised counts were used for cluster identification for all genes except for *CD4* and *B3GAT1* (where, due to low gene capture [65] in all clusters, means were used). Clusters without expression of any of *CD3D*, *CD3E*, *CD3G* were excluded to ensure only T-cells were analysed. Double negative clusters were defined by negative 75th percentile expression of *CD8A* and *CD8B*, and negative mean expression of *CD4*. Double positive clusters were defined by positive expression of these genes. The CD4^+^ T-cells were defined as the remaining clusters with positive mean expression of *CD4*. The CD8^+^ T-cells were defined as the remaining clusters which co-express *CD8A* and *CD8B*. The following filters were used for cluster definitions of validated cell populations: Tregs (positive expression of *FOXP3*); Senescent (negative expression of *CD27* and *CD28*, positive 75th percentile expression of *KLRG1* and positive mean expression of *B3GAT1*); Exhausted (positive 75th quantile expression of ≥4 of *HAVCR2*, *PDCD1*, *TOX*, *LAG3*, *CTLA4*, *TIGIT*, *CD38*, *ENTPD-1* and positive expression of TRDC was filtered out to exclude gamma-delta T-cells).

### 5.13. Single-Cell RNA Sequencing Analysis: Data Analysis and Figures

Differential expression analysis was performed with the FindMarkers function in Seurat. Scaled expression was extracted from Seurat tumour samples only and truncated violins were plotted with Prism (GraphPad). Normalised expression heatmaps were plotted in Prism (GraphPad) using matrices of 75th percentile and mean expression.

## Figures and Tables

**Figure 1 cancers-13-01776-f001:**
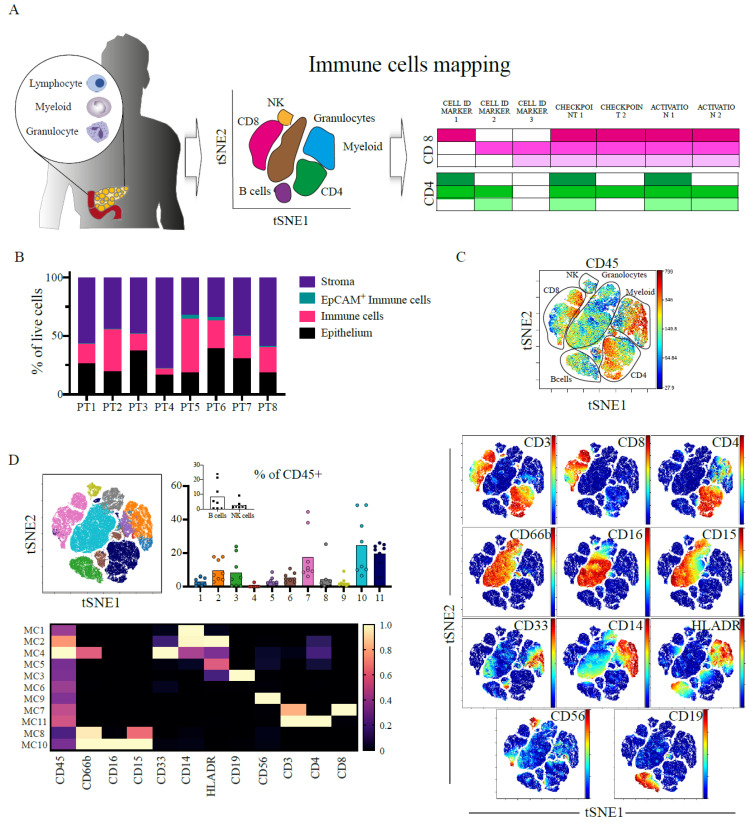
Immune infiltration into PDAC is heterogenous but with a marked T-cell population. (**A**) Schematics of the experimental procedure; primary resectable pancreatic tumours are made into single-cell suspensions and taken for phenotyping using mass cytometry (CyTOF). CyTOF data was clustered using cytobank FlowSOM to identify common populations across patients. Using a set of lineage markers, checkpoints and activation markers cellular states and functionality are defined with a focus on T-cell populations. (**B**) 200,000 cells pooled from 8 patients were gated in silico and cellular granularity was assessed; Immune cells (CD45^+^), epithelial cells (EpCAM^+^), Stroma (CD45^-^EpCAM^−^) and EpCAM^+^ Immune cells (CD45^+^ EpCAM^+^). (**C**) 100,000 CD45^+^ cells pooled from 8 patients and viSNE analysis using main cell lineage markers was performed to identify the main immune cell populations. viSNE plot is shown with manual annotation of cell identities (top), expression profile of the CyTOF markers used for clustering is shown (bottom). (**D**) viSNE plot of the main immune populations coloured and labelled by FlowSOM. Bar plots of metacluster frequencies in each patient. Inset shows the B-cell and NK clusters (CV % 111.3 and 108.6% respectively). Heatmap of FlowSOM metaclusters of CD45^+^ cells; rows represent metaclusters from combined single cells across patients. (**E**) NK cells were clustered with FlowSOM and 7 different metaclusters identified. Metacluster’s relative frequency is presented in the bar plot. Inset shows the lower frequency metaclusters. Expression profile for each metacluster is shown in the heatmap (right). The major metacluster being a CD8^+^ NK population. (**F**) Granulocyte were clustered with FlowSOM and 7 different metaclusters identified. Metacluster’s relative frequency is presented in the bar plot. Inset shows the lower frequency metaclusters. Expression profile for each metacluster is shown in the heatmap (right). The major metacluster expressing an intermediate level of CD16 and CD15. (**G**) Myeloid cells were clustered with FlowSOM and 6 different metaclusters identified. Metacluster’s relative frequency is presented in the bar plot. Inset shows the lower frequency metaclusters. Expression profile for each metacluster is shown in the heatmap (right). The major metacluster expressing an intermediate level of CD14 and CD33 but high for MHCII (HLA-DR), and another important metacluster is the one lacking HLA-DR expression (MSDC). All bar plots are median and the individual dots are individual patients. Heatmaps are normalised for each marker with lowest expression marked in dark blue as zero, and highest in yellow as 1. MC= metacluster.

**Figure 2 cancers-13-01776-f002:**
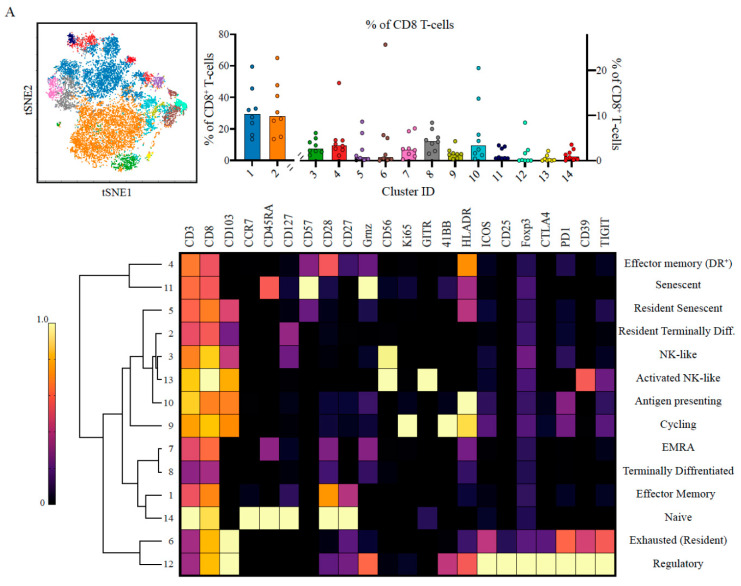
CD8^+^ T-cell senescence and activated Tregs dominate the landscape of the tumour. (**A**) ~14,750 CD8^+^ T-cells were pooled and 14 metaclusters identified with FlowSOM and visualised on viSNE plot. Metaclusters’ relative abundance is shown in the bar plot MC1,2 are plotted on the left *y*-axis, MC3-14 are plotted on the right *y*-axis. The heatmaps show the expression profile of immune checkpoints in the different metaclusters and identified unique populations based on the combinatorial expression. Note that the major CD8^+^ populations are effector memory cells and the presence of 4 metaclusters corresponding to senescent cells. There is also a proportion of exhausted cells as well as metaclusters of activated cells. (**B**) ~17,870 CD4^+^ T-cells were pooled and 10 metaclusters identified with FlowSOM and visualised on viSNE plot. Metaclusters’ relative abundance is shown in the bar plot. The heatmaps show the expression profile of immune checkpoints in the different metaclusters and identified unique populations based on the combinatorial expression. The major populations are central memory cells, and there were 5 metaclusters identified as regulatory T-cell (analysed in depth in (**C**)). (**C**) ~3900 CD4^+^ regulatory T-cells were pooled and 8 metaclusters identified with FlowSOM and visualised on a viSNE plot. Metaclusters’ relative abundance is shown in the bar plot. The heatmaps show the expression profile of immune checkpoints in the different metaclusters and identified unique populations based on the combinatorial expression. More than 50% of the metaclusters show an activated phenotype albeit at different magnitudes. The subpopulations are annotated based on the combination of checkpoints and their level of expression. All bar plots are median and the individual dots are individual patients. Heatmaps are normalised for each marker with lowest expression marked in dark blue as zero, and highest in yellow as 1. Hierarchal clustering of heatmaps was done in Morpheus.

**Figure 3 cancers-13-01776-f003:**
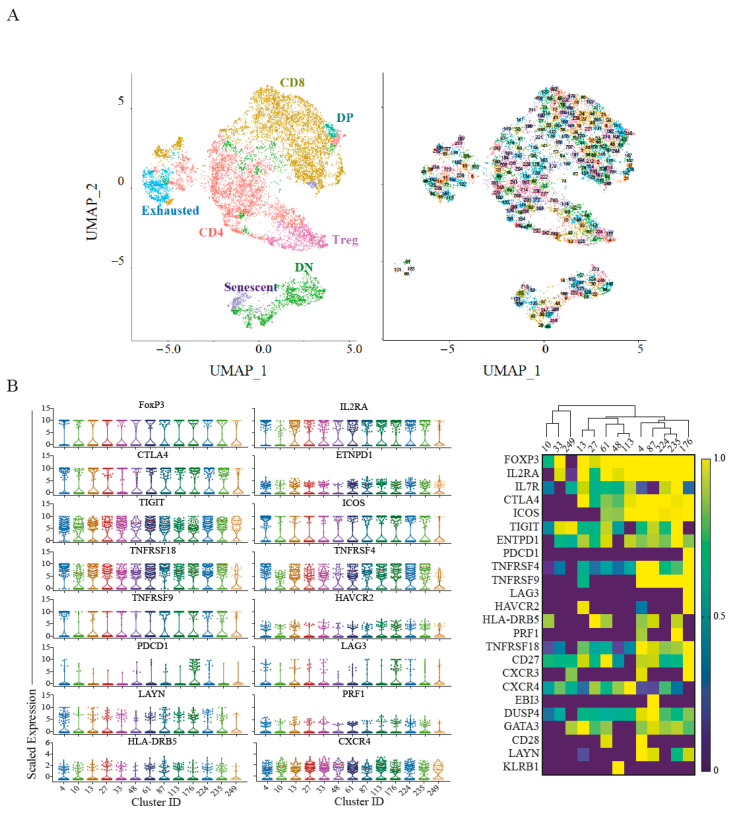
Single Cell RNA sequencing reveals senescence and regulatory signature of T-cells. (**A**) Left, ~13,600 T cells in tumour are projected on UMAP and clusters annotated by the major compartments including regulatory T-cell (pink), senescent T-cells (purple) and exhausted T-cells (cyan). Right, UMAP of the cluster identities in each of the major subsets. (**B**) Left, Violin plots depicting the 75th percentile scaled expression of the marker genes in the Treg clusters. Right, heatmaps showing the top differentially expressed genes in the Treg clusters. Expression has been normalised per gene. (**C**) Left, Violin plots of the 75th percentile expression of the key gene signature for the senescent population. Right, Heatmaps showing NK and senescence genes uniquely expressed in the senescent population. Heatmap scale for *B3GAT1* is presented as mean values per cluster rather than the 75th percentile due to low capture of this gene. (**D**) Left, Violin plots of the 75th percentile expression of the key gene signatures for the exhausted T-cell population. Right, the corresponding heatmaps of the key gene signatures. Full expression profiles per cluster are provided in Appendix A.

**Figure 4 cancers-13-01776-f004:**
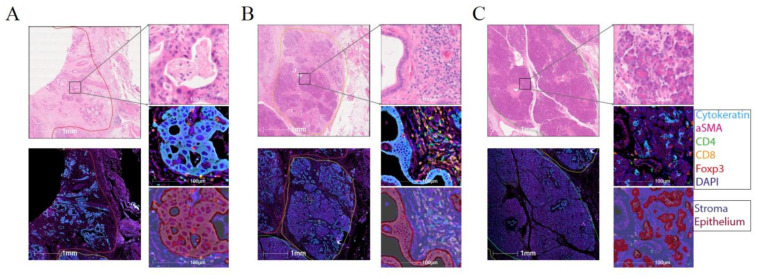
Multiplex imaging of T-cell distribution within PDAC tumours identifies a stroma restricted Treg compartment. (**A**) 2× magnification (left) of a cancer region showing H&E (top) and immunofluorescence (bottom). 40× magnification (right) of a region from (left) showing H&E staining (top), the fluorescence signal (middle) and the region classification into epithelium and stroma (bottom). (**B**) As in (**A**) for a region exhibiting pancreatitits. (**C**) As in (**A**) for a normal pancreas region. (**D**) Infiltration of CD8^+^ T-cells (left), CD4^+^ T-cells (middle) and Tregs (right) into the tissue (Cancer, Pancreatitis and Normal), as well as sub-tissue architectural distribution between the epithelium-rich and stroma-rich areas. Mixed-effect ANOVA with Tukey’s correction. (**E**) Proximity analysis of CD8^+^ T-cells distance distribution within 50 µm of epithelial cells fitted using a lognormal distribution with geometric means and R^2^ as follows: Cancer: 17.21 µm, 0.80; Pancreatitis 14.55 µm, 0.79; Pancreas: 8.46 µm, 0.99. (**F)** Proximity analysis of Treg distance distribution within 50 µm of CD8^+^ T-cells, fitted using a lognormal distribution with geometric means and R^2^ as follows: Cancer: 16.75 µm, 0.84; Pancreatitis 14.28 µm, 0.82; Pancreas: 11.24 µm, 0.92. Scale bars: 1 mm (for 2× magnification) and 100 mm (for 40× magnification).

## Data Availability

Data available upon reasonable request from authors.

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
