# Peer review of "Activated Regulatory T-Cells, Dysfunctional and Senescent T-Cells Hinder the Immunity in Pancreatic Cancer"

_cancers, 2021, doi:10.3390/cancers13081776_

Round 1

Reviewer 1 Report

The article entitled: "Activated regulatory T-cells, dysfunctional and senescent T-2 cells hinder the immunity in pancreatic cancer" by Sivakumar et al. characterizes the phenotype of different T-cell population that infiltrate tumors, which correlates with the immunosuppressive microenvironment of pancreatic cancer. The manuscript is overall well written and results provide a rationale for evaluate other immunocheckpoint inhibitors in further clinical trials. The introduction is poor especially to TME and its different cell population, and the failed results with immunocheckpoint inhibitors in PDAC. Therefore, the article need some amendments before be considered for publication. Please find below my comments:

1.-Supplement introduction with more information about TME and previous results using immunocheckpoint inhibitors. For this purpose, it may be useful the following articles: Martinez-Useros J. et al., Cancers (Basel). 2021 Jan 17;13(2):322; Krishnamoorthy M., et al., Cancers (Basel). 2020 Nov 12;12(11):3340 and Mahalingam D., et al. Cancers (Basel). 2018 May 25;10(6):160.

2.- Figure 1 must be placed in before Figure 2.

3.-Please include a Table with clinico-pathological characteristics of patients recruited for the study.

4.- Figure 1B shows different percentage of cell population between 8 patients; is some cell population enrichment associated to any clinico-pathologic characteristic of patients?

5.-The article must be formatted, 2.1 section has not numeration and an appropriate heading and 2.1 is actually 2.2.

6.-Please include at the end of each sub-heading of the result section a 3-5 lines summary-conclusion of the finding, similarly to those in section 2.2.

7.-Figure 2 A. Please include units in both axes of the bar-graph.

Reviewer 2 Report

Sivakumar et al. investigate the composition of immune infiltrate in primary pancreatic ductal adenocarcinomas (PDAC) via analysis by mass-cytometry, multiplex immunohistochemistry and single cell RNAseq using a published dataset. This is an interesting descriptive study that adds to the growing collection of immune markers in human PDAC, and the authors have added an intriguing observation that expression of checkpoint molecules other than PD-1 may facilitate immunosuppression. Overall, while the study analyzes only a few samples, the authors do identify multiple potential targets on T cells that could be validated in follow up studies and may serve as new therapeutic vulnerabilities. Minor points. 1. Patient status with regards to treatment with antibiotics could be informative. 2. Figures are out of order 3. Histology/IF images in Figure 4 are difficult to see

Round 2

Reviewer 1 Report

Thanks all authors for the ammendments in the manuscript. The article has been improved accordingly.